# Effect of Chest Physiotherapy on Quality of Life, Exercise Capacity and Pulmonary Function in Patients with Idiopathic Pulmonary Fibrosis: A Systematic Review and Meta-Analysis

**DOI:** 10.3390/healthcare11222925

**Published:** 2023-11-08

**Authors:** Javier Martín-Núñez, Alejandro Heredia-Ciuró, Laura López-López, Andrés Calvache-Mateo, Sofía Hernández-Hernández, Geraldine Valenza-Peña, Marie Carmen Valenza

**Affiliations:** Department of Physiotherapy, Faculty of Health Sciences, University of Granada, Av. De la Ilustración, 6018016 Granada, Spain; javimartinn29@gmail.com (J.M.-N.); ahc@ugr.es (A.H.-C.); lauralopez@ugr.es (L.L.-L.); andrescalvache@ugr.es (A.C.-M.); valenzagera@gmail.com (G.V.-P.)

**Keywords:** idiopathic pulmonary fibrosis, chest physiotherapy, quality of life, exercise capacity, pulmonary function

## Abstract

Background: Idiopathic pulmonary fibrosis (IPF) is one of the most deleterious diseases of the pulmonary fibrosis spectrum. Its clinical presentation includes irreversible loss of lung function, and increasing cough, dyspnea and impaired quality of life. Chest physiotherapy can improve ventilation capacity, gas exchange, dyspnea, exercise capacity and quality of life. The aim of this study was to review the evidence about chest physiotherapy in IPF, specifically meta-analyzing quality of life, exercise capacity and pulmonary function. Methods: A wide search was conducted in PubMed, Embase, Cochrane and Web of Science for articles included until October 2023. PROSPERO Identifier: CRD42022333496. The Downs and Black scale and the Cochrane tool were employed to evaluate quality assessments and to assess the risk of bias. Data were pooled, and a meta-analysis was conducted. Results: We selected 10 studies in which a chest physiotherapy program was performed with a total of 340 patients; of these, three articles were meta-analyzed. Significant effects in favor of chest physiotherapy were found for quality of life (MD = −8.60, 95% CI = −11.30, −5.90; *p* < 0.00001; I_2_ = 24%), exercise capacity (MD = 37.62, 95% CI = 15.10, 60.13; *p* = 0.001; I_2_ = 65%) and pulmonary function (MD = 7.86, 95% CI = 2.54, 13.17; *p* = 0.004; I_2_ = 80%). Conclusions: The systematic review showed significant results for the application of chest physiotherapy regarding pulmonary capacity, diffusion of gases and quality of life in IPF patients. The meta-analysis showed a significant improvement associated with applying chest physiotherapy in pulmonary function, exercise capacity and quality of life.

## 1. Introduction

Pulmonary fibrosis results from a variety of insults to the lung that include toxic, autoimmune, drug-induced, infectious or traumatic injuries leading to a dysfunction of the normal structure and physiology of alveolar epithelial tissues [1]. In this vein, pulmonary fibrosis is considered as one of the possible tissue responses to injury [2]. Of the 150 pathologies related to pulmonary fibrosis, idiopathic pulmonary fibrosis (IPF) is the most common and one of the most deleterious ones [3]. Specifically, IPF has a high death rate in individuals aged 3 to 5 years [4] and a higher incidence in males than females, showing a ratio of 2:1 [5,6,7]. Additionally, the prognosis and estimated survival rate are related to some factors including old age; smoking habit; low body mass index; impaired pulmonary function; and the presence of comorbidities, particularly pulmonary hypertension and emphysema [8,9,10,11,12]. The clinical presentation of IPF is characterized by irreversible loss of lung function, manifested as symptoms of cough and dyspnea, and impaired quality of life [13]. Specifically, dyspnea results from an increased load of inspiratory muscles, a functional weakness of respiratory muscles, an imbalance between ventilatory demand and capacity, and a dysfunction of gas exchange [14].

Evidence on the clinical management of IPF is rapidly evolving and always includes drug therapy as a first-line treatment. While no pharmacological therapy has been universally accepted to modify the disease course of IPF, there are different mechanism-based anti-fibrotic therapies proven by large RCTs to modify disease progression [15]. In another line, other novel therapeutic targets have emerged accompanied by other supportive therapies. Supportive therapies include pulmonary rehabilitation, management of comorbidities and education programs accompanied by supplemental oxygen if necessary [16]. Pulmonary rehabilitation (PR) is defined as a comprehensive intervention based on thorough patient assessment followed by patient-tailored therapies that include, but are not limited to, exercise training, education and behavior change. PR in IPF traditionally includes the use of chest physiotherapy, which can improve ventilation capacity, gas exchange and the function of respiratory muscles, reducing the dynamic hyperinflation added to the dyspnea, and improving the exercise capacity and quality of life [17,18].

Overall, owing to diagnostic challenges, updated diagnostic criteria and differences in treatment methodologies, there is substantial heterogeneity between studies estimating the effectiveness of chest physiotherapy in IPF [4,12]. To date, neither the concrete use of chest physiotherapy nor the organizational aspects of pulmonary rehabilitation programs in IPF have been analyzed. Considering this, we decided to systematically review the evidence about chest physiotherapy in IPF patients, specifically regarding quality of life, exercise capacity and pulmonary function, and to perform a meta-analysis of these variables.

## 2. Methods

### 2.1. Study Registration

Our systematic review was registered with PROSPERO (International Prospective Register of Systematic Reviews) under the registration number CRD42022333496 and with publicly available criteria for this review. We adhered this systematic review to the guidelines outlined in the Preferred Reporting Items for Systematic Reviews and Meta-analyses (PRISMA) [19] and followed the Cochrane Collaboration’s handbook guidelines for systematic reviewing interventions [20].

### 2.2. Search Strategy

We systematically searched for articles indexed in the PubMed, Cochrane Library, Embase and Web of Science databases from their inception to October 2023. The search strategy was developed by combining the use of the terms included in the Medical Subject Headings (MeSH) and keywords. The search strategy underwent testing and refinement to ensure it was the most effective approach for this review, considering (1) participants (“idiopathic pulmonary fibrosis” OR “IPF”) and (2) interventions (“pulmonary rehabilitation” OR “rehabilitation” OR “respiratory rehabilitation” OR “rehabilitation training” OR “rehabilitation program” OR “rehabilitation therapy” OR “physiotherapy” OR “physical therapy” OR “breathing technique” OR “resistance training” OR “physical training” OR “exercise” OR “exercise therapy” OR “breathing exercise” OR “exercise training” OR “exercise program” OR “chest” OR “chest physiotherapy”). We screened the references of relevant reviews to find additional studies that could potentially be included in this review. (See Appendix A).

Articles were included by applying the following eligibility criteria of the PICOS model [20]:

(1) Patients had to be adults with a diagnosis of idiopathic pulmonary fibrosis in accordance with the clinical guidelines of the American Thoracic Society (ATS) and the European Respiratory Society (ERS);

(2) Interventions had to include chest physiotherapy, as described by Warnok et al. (i.e., conventional chest physiotherapy, positive expiratory pressure mask therapy, high-pressure PEP mask therapy, active cycle of breathing techniques, autogenic drainage, exercise and oscillating devices) [21] isolated or in combination with other techniques;

(3) The comparator group had to be standard medical care or programs without chest physiotherapy;

(4) Outcomes had to include respiratory function, exercise capacity and/or quality of life;

(5) Eligible studies had to be randomized controlled trials, quasi-experimental trials or pilot studies.

Only articles in English, Spanish or French were included. Articles were excluded if they were (1) abstracts presented in congresses or protocols, or (2) duplicated publications.

Two researchers performed the search process, which included removing the duplicates and screening the titles, abstracts and eligible full texts. Additionally, the researchers independently performed the literature search, and disagreements were resolved through a consensus discussion with a third independent investigator to reduce the selection bias potential. Data extraction was performed using files with predefined categories that included the reference, participant characteristics (i.e., age, gender, severity, etc.), interventions (i.e., type of chest physiotherapy, duration, frequency, etc.), intervention outcomes (variables measured and outcomes) and follow-up.

Data extraction and quality assessment were conducted after the articles had been chosen. The methodological quality of the studies was evaluated using the Downs and Black quality assessment method [22]. This method includes 27 items with five subscales (i.e., study quality, external validity, study bias, confounding and selection bias, and study power). The study is considered “excellent” if it has a score of 26 or higher, “good” if it ranges between 20 and 25, “fair” if it ranges between 15 and 19 and “poor” if its score is 14 or lower. This scale is categorized as one of the six highest-quality assessment scales for using in systematic reviews, due to its high validity and reliability [23,24].

Risk of bias was assessed with the Cochrane Collaboration Risk of Bias Tool for Randomized Controlled Trials (ROB2) [19]. This tool comprises five elements susceptible to bias and categorizes the quality assessment as follows: poor quality, when significant limitations are present that could invalidate the results and when two or more criteria are listed as having high or unclear risk of bias; fair quality, if one criteria is unmet (i.e., high risk of bias in one domain) or two criteria remain unclear, but these do not present limitation that could invalidate the results; or high quality, when all domain exhibit low risk [25]. In studies in which a non-randomized experimental intervention was performed, we used the Risk of Bias in Non-randomized Studies of Interventions scale (ROBINS-I) [26], which consists of seven elements that can be used to assess the risk of bias. Studies are considered to have low risk of bias when all domains show low risk, moderate risk when all domains show low or moderate risk of bias, serious risk when serious risk of bias is shown in at least one domain and critical risk when critical risk of bias is shown in at least one domain. Two researchers assessed the methodological quality of the studies and risk of bias separately, and a third researcher intervened if there was any disagreement.

The qualitative analysis was based on classifying the results into levels of evidence according to the Grading of Recommendations, Assessment, Development and Evaluation (Grade), which is based on five domains: study design, imprecision, indirectness, inconsistency and publication bias [27].

The evidence has been divided into the following four levels as follows: (a) very low quality, reflecting a situation where any estimate of the effect is highly uncertain, with three of the five domains not being met; (b) low quality, where further research is very likely to have a substantial impact on our confidence in the estimate of the effect and is likely to result in changes, given that two of the five domains are not met; (c) moderate quality, indicating that additional research is expected to have a significant impact on our confidence in the estimate of the effect, and it might lead to changes in the estimate due to one of five domains not being met; and (d) high quality, where further research is highly unlikely to alter our confidence in the estimate of the effect, and all five domains are fully met [28,29].

The assessment of the five domains was carried out following the criteria set by GRADE. In terms of the study design domain, recommendations were lowered by one level if there existed uncertainty or a high risk of bias, coupled with significant limitations in estimating the effect. Concerning inconsistency, recommendations were also downgraded by one level when point estimates exhibited wide variation among studies, confidence intervals showed minimal overlap or I_2_ indicated substantial or large heterogeneity. Within the domain of indirectness, recommendations faced downgrading when there were substantial disparities in interventions, study populations or outcomes. In relation to the imprecision domain, recommendations were downgraded by one level if there were fewer than 400 participants for continuous data.

### 2.3. Meta-Analysis

Review Manager 5 (Rev-Man version 5.1, update March 2011) software was used to perform the quantitative synthesis of all studies that presented the post-intervention means and standard deviations of quality of life (SGRQ), exercise capacity (6MWT) and pulmonary function (Dl_co_). Quantitative data, which included the number of patients evaluated, final mean values and standard deviations for each treatment group, were extracted to calculate overall mean differences between the experimental and control arms.

When the studies did not present sufficient data to calculate the effect size (e.g., missing means or standard deviations), we contacted the author to obtain the necessary information. In cases where *p* values or 95% confidence intervals were available but standard deviations were missing, we calculated them using Review Manager’s integrated calculator.

For the analysis of each scale, standardized mean differences were used. Overall mean effect sizes were determined using either random-effects models or fixed-effects models, depending on the results of the I^2^ statistical heterogeneity test (fixed-effects models were used for I_2_ values below 50%) [19]. In addition, forest plots were visually inspected to identify possible outlier studies. Additionally, we performed a visual examination of the forest plots to identify outlier studies, explored potential sources of heterogeneity and performed a sensitivity analysis by excluding trials whose weight in the analysis was too high and may be biasing the statistically significant difference in favor of the experimental group.

## 3. Results

### 3.1. Study Selection

A flow diagram of the search, screening and selection process is shown in Figure 1. Ten studies [30,31,32,33,34,35,36,37,38,39] were included in this systematic review. Of these, four were randomized controlled trials [31,32,33,34], five were quasi-experimental trials [30,32,34,35,36], and one was a pilot study [33].

### 3.2. Study Characteristics

The characteristics and quality of the studies are shown on Table 1. A total of 340 patients with idiopathic pulmonary fibrosis received chest physiotherapy. The sex distribution was heterogeneous, ranging between 5.12 and 41.9% of women. Mean age ranged in the different groups from 54.4 ± 6.1 to 68.8 ± 6 years old.

Pulmonary function was assessed with the diffusing capacity of the lungs for carbon monoxide (Dl_co_). The Dl_co_ showed higher heterogeneity, with values between 38 ± 13 and 68 ± 32.3. Heterogeneity was also observed in the location of the studies, which were mainly performed in European and Asian countries, and the clinical site where most interventions were performed was a hospice. As regards the existence of a follow-up assessment, it was only performed in five studies [31,32,33,34,38], ranging from one week to one year. In addition, only four studies received funding for their implementation [30,31,32,33].

### 3.3. Risk of Bias in Studies

The methodological quality of the studies was fair, with results ranging from 13 to 24 when the Downs and Black quality tool was applied. When the ROB was applied to randomized controlled trials, three of them [31,32,34] showed low risk and only the study by Jarosch et al. [33] showed fair quality. As regards the ROBINS-I for non-randomized controlled trials, three studies showed low risk of bias [30,37,38] and three showed serious risk of bias [35,36,39] (see Appendix A).

### 3.4. Results of Individual Studies

The description of the interventions conducted and the results of the studies are shown on Table 2. The chest physiotherapy programs of the studies were heterogeneous in their type and form of application. The most frequently applied types of chest physiotherapy were breathing techniques, particularly pursed-lipped, diaphragmatic breathing, and control deep breathing. However, two studies additionally included conventional chest physiotherapy [30,33].

Most of the studies included a combination of strength and endurance exercises [30,33,37,38]. Only two studies included endurance exercises in isolation [34,36]. Five studies also included aerobic training [30,33,34,36,39]. Additionally, some of the studies included educational programs [33,34,36,37,38].

Three studies applied the usual care of IPF to the control groups [32,33,34], while other two studies proposed recommendations about physical activity [30,31].

As regards the form of application, the duration of the intervention ranged from 3 to 48 weeks. The frequency of the treatment ranged between 2 and 7 days per week and 15–120 min per session. The chest physiotherapy programs were supervised by healthcare professionals, except in the study by Ozalevli et al. [39], which included a non-supervised intervention, and the study by Shen L et al. [32], which combined a supervised program with a non-supervised home program.

The most outcomes frequently assessed were quality of life, exercise capacity and pulmonary function. The quality of life of the chest physiotherapy group improved in most of the studies [30,31,32,33,34,35,37,39]. As regards exercise capacity, most studies [30,31,32,33,34,35,37,38,39] showed a significant improvement. Finally, pulmonary function improved in all the studies that evaluated it, but only the study by Shen L et al. [32] showed significant differences compared to baseline and the control group.

According to the GRADE recommendations, there was general low-quality evidence regarding the effects of chest physiotherapy on pulmonary function, exercise capacity or quality of life, being downgraded due to study design, risk of bias (performance and detection bias), imprecision due to sample size and inconsistency (range I_2_ = 24 to 80%). Moderate-quality evidence was found regarding the isolated effects of chest physiotherapy on all variables, being downgraded due to imprecision (n = 82). In addition, there was low-quality evidence regarding the effects of chest physiotherapy regarding pulmonary function, being downgraded due to risk of bias (performance and detection bias), inconsistency (I_2_ = 80%) and imprecision (n = 340). Finally, there was moderate-quality evidence regarding the effects of both chest physiotherapy combined with therapeutic exercise, being downgraded due to risk of bias (selection, performance and detection bias) and inconsistency (I_2_ = 54%).

### 3.5. Results of Syntheses

The results obtained in the meta-analysis were analyzed comparing the chest physiotherapy groups to the usual care groups.

The results obtained on quality of life were analyzed as shown in Figure 2.

The pooled mean difference showed a significant overall effect of chest physiotherapy interventions in the experimental group compared to the control group (MD = −8.60, 95% CI = −11.30, −5.90; *p* < 0.00001). The results showed low heterogeneity and a significant variability of I_2_ = 24% not attributable to chance.

The results obtained for exercise capacity were analyzed as shown in Figure 3. The pooled mean difference showed a significant overall effect of chest physiotherapy interventions in the experimental group compared to the control group (MD = 37.62, 95% CI = 15.10, 60.13; *p* = 0.001). The results showed heterogeneity and a variability of I_2_ = 65% not attributable to chance. Because of the high heterogeneity, the sensitivity meta-analysis was carried out excluding the study by Zhou et al. (1) [31] due to its weight in the analysis. However, the statistically significant difference in favor of the experimental group was maintained (MD = 49.61, 95% CI = 32.84, 66.38; *p* < 0.00001)

The results obtained for pulmonary function were analyzed as shown in Figure 4. The pooled mean difference showed a significant overall effect of chest physiotherapy interventions in the experimental group compared to the control group (MD = 7.86, 95% CI = 2.54,13.17; *p* = 0.004). The results showed heterogeneity and a variability of I_2_ = 80% not attributable to chance. Because of the high heterogeneity, the sensitivity meta-analysis was carried out excluding the study by Zhou et al. (1) [31] due to its weight in the analysis. However, the statistically significant difference in favor of the experimental group was maintained (MD = 9.36, 95%, CI = 1.75,16.98; *p* = 0.02).

## 4. Discussion

The aim of this review was to evaluate the effects of chest physiotherapy programs in idiopathic pulmonary fibrosis patients. Our results showed that the application of chest physiotherapy programs led to significant improvements in pulmonary function, exercise capacity and quality of life of patients with idiopathic pulmonary fibrosis.

Although this article is the first systematic review about the effects of chest physiotherapy programs in IPF patients, various scientific guidelines have recommended pulmonary rehabilitation for this pathology with different levels of evidence.

Routine chest physiotherapy programs [40] included instrumental techniques and manual/active physiotherapy techniques, with a reported physiological effect of assistance in the clearance of respiratory secretions and improved breathing. In this line, these techniques are usually included on pulmonary rehabilitation programs.

Unfortunately, although several reviews have been made of the short- and long-term effects of pulmonary rehabilitation on IPF, the results remain controversial when they do not analyze the main components of the programs. Additionally, our review aimed to analyze the GRADE value for chest physiotherapy when used for IPF patients, and while our results show moderate recommendation, this is the first review on IPF to use it. The review by Kenn et al. [41] showed that pulmonary rehabilitation was able to improve exercise capacity and health-related quality of life in IPF patients in the short term. However, another review [42] concluded that the positive effects of rehabilitation remain elusive. Except for those recent reviews, none of the previous reviews have analyzed the use and effectiveness of a relevant treatment modality of pulmonary rehabilitation such as chest physiotherapy in IPF.

Various therapeutic interventions were analyzed considering their effects on health status and reducing morbidity–mortality. For example, guided physical activity has been recommended by the World Health Organization, the American Centers for Disease Control and Prevention and the American College of Sport Medicine [41,43,44,45,46]. Thus, the recommendation for chest physiotherapy has never been explored but is generally perceived to be an appropriate treatment for IPF patients, even if the expected benefits are small, i.e., a “weak” recommendation. 

Due to the chronicity and difficult management of IPF, health-related quality of life is an important endpoint in research and clinical practice evaluation. In this regard, our analysis showed that chest physiotherapy was able to improve health-related quality of life when compared to educational, pharmacological and guided physical activity treatments.

Moreover, previous published research has shown clinical improvements after supervised exercise for pulmonary fibrosis, with significant improvements in daily life [34,47,48,49].

Exercise capacity has been regarded as a predictor of survival in IPF patients [50] and included in numerous trials about pulmonary rehabilitation programs in IPF. In this review, we showed that exercise capacity is usually included as a primary outcome in programs involving chest physiotherapy. Our analysis showed that chest physiotherapy was able to improve exercise capacity significantly when compared to other programs.

Previous reviews [47,51,52] have reported a high efficacy of pulmonary rehabilitation in dyspnea, exercise capacity and quality of life, similarly to our review. However, chest physiotherapy showed improvements in other variables such as pulmonary function.

These results should be considered for their clinical relevance, given the clinical evolution of idiopathic pulmonary fibrosis patients, since their limitations affect exercise capacity, reported symptoms and quality of life [53]. Additionally, according to the GRADE recommendation, the chest physiotherapy shows moderate recommendation when considering pulmonary function, exercise capacity and quality of life.

Additionally, previous studies have reported significant improvements in psycho-emotional aspects such as anxiety and depression [54]. In this regard, our results showed improvements in psycho-emotional status in studies that evaluated it. The chest physiotherapy also showed significant improvements in other chronic respiratory diseases [55].

## 5. Limitations

This review has several limitations to consider. First, the heterogeneity of the study design may have decreased the quality of the study. The high presence of prospective non-randomized controlled trials reduced the sample size of the meta-analysis. Future randomized controlled trials are necessary to improve the quality of the quantitative analysis, adding additional variables that are underexplored. However, other reviews have been performed with a similar study sample [49]. Moreover, the multidimensional nature of the intervention programs made it difficult to study the effect of chest physiotherapy in isolation. Finally, while we conducted a comprehensive review of various electronic databases encompassing both published and unpublished studies, it is possible that certain studies eluded our search.

## 6. Conclusions

The results of the systematic review showed significant results of the application of chest physiotherapy programs regarding pulmonary capacity, diffusion of gases and quality of life in patients with idiopathic pulmonary fibrosis. The meta-analysis showed a significant positive improvement resulting from applying chest physiotherapy programs in pulmonary function, exercise capacity and quality of life. The addition of chest physiotherapy in pulmonary rehabilitation must be analyzed considering the possible physiologic effects and the modality of the chest physiotherapy included. In this line, more studies have to be performed on instrumental breathing techniques to analyze their results.

These results should be taken with caution. The heterogeneity of the scientific evidence design limited the quality of the results due to the small number of studies included in the quantitative analysis. It is necessary to conduct future randomized controlled trials analyzing more chest physiotherapy programs. However, the RoB and Downs and Black tools showed that the results of this review presented strong evidence for the application of chest physiotherapy in IPF. For these reasons, it is consistent to suggest including chest physiotherapy programs, with a focus on breathing techniques, in the clinical approach of patients with idiopathic pulmonary fibrosis.

In another line, the GRADE recommendation for chest physiotherapy shows moderate recommendation when considering pulmonary function, exercise capacity and quality of life due to the heterogeneity of the included studies.

## Figures and Tables

**Figure 1 healthcare-11-02925-f001:**
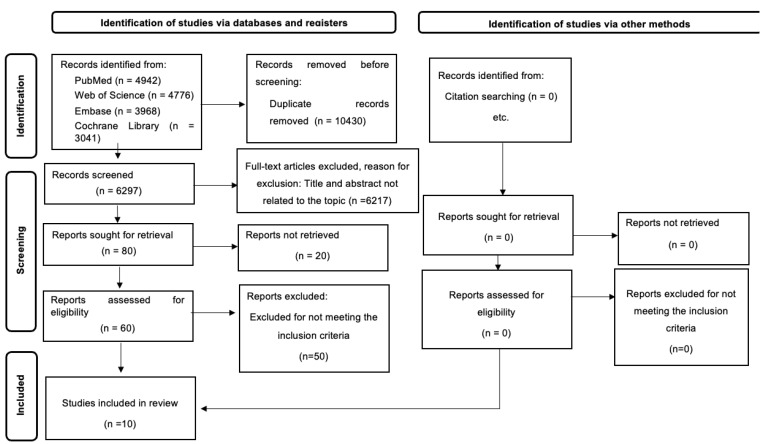
Flow chart of the selected studies.

**Figure 2 healthcare-11-02925-f002:**
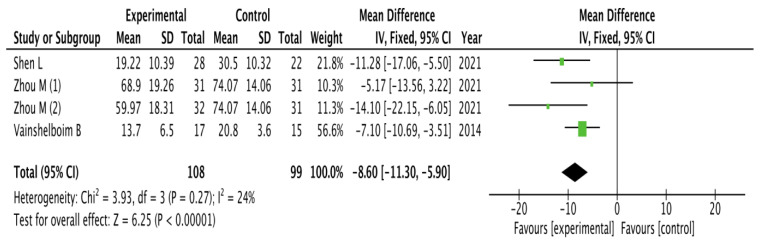
Results of Quality of life [31,32,34].

**Figure 3 healthcare-11-02925-f003:**
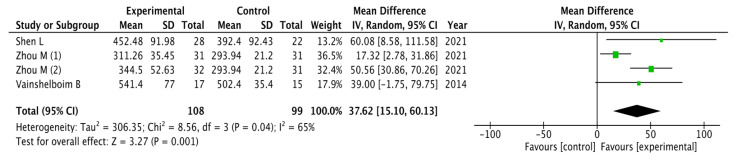
Results of Exercise capacity [31,32,34].

**Figure 4 healthcare-11-02925-f004:**
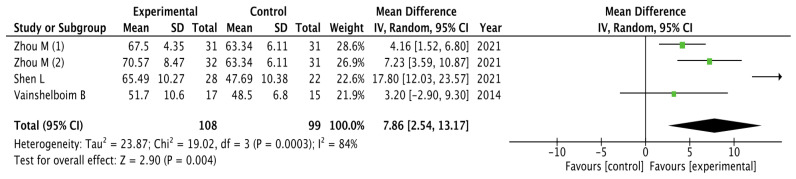
Results of Pulmonary function [31,32,34].

**Table 1 healthcare-11-02925-t001:** Characteristics and quality of the included studies.

Study (Year)	Location	Study Design: Groups	N Groups (%Women)	Age (Years ± SD)	Lung Function (FVC (L or %))	Follow-Up	Quality of the Study (Risk of Bias)
Choi HE et al. (2023) [30]	South Korea Hospital	Prospective Non-RCT 2 Groups	G1: 13 (15) G2: 12 (0)	G1: 68 ± 5.3 G2: 69 ± 5.9	G1: 78.3% G2: 83.2%	-	18/28 (Low)
Zhou M et al. (2021) [31]	China Hospital	RCT 3 Groups	G1: 31(41.94) G2: 32 (40.62) G3: 32 (45.16)	G1: 64 ± 9 G2: 66 ± 11 G3: 67 ± 10	G1: 2.25 ± 0.70 G2: 2.17 ± 0.49 G3: 2.12 ± 0.49	6 months	23/28 (Low)
Shen L et al. (2021) [32]	Shanghai Hospital	RCT 2 Groups	G1: 39 (5.12) G2: 43 (6.97)	G1: 65.31 ± 6.11 G2: 64.95 ± 7.97	G1: 2.50 ± 0.59 G2: 2.45 ± 0.8	1 year	23/28 (Low)
Jarosch I et al. (2020) [33]	Germany University and Medical School	RCT 2 Groups	G1: 34 (24) G2: 17 (19)	G1: 68 ± 9 G2: 65 ± 10	G1: 74% ± 19% G2: 72% ± 17%	3 months	24/28 (Some Concerns)
Vainshelboim B et al. (2014) [34]	Israel Medical Center	RCT 2 Groups	G1: 15 (33) G2: 17 (35)	G1: 68.8 ± 6 G2: 66 ± 9	G1: 66.1% ± 14.8% G2: 70.1% ± 17.4%	1 week	18/28 (Low)
Rifaat N et al. (2014) [35]	El-Minya Hospital	Prospective Non-RCT 1 Group	G: 30 (73.3)	G: 54.4 ± 6.1	G: 51.9% ± 13.6%	-	13/28 (High)
Swigris J et al. (2011) [36]	Denver Metropolitan Area	Pilot study 1 Group	G: 21 (14.2)	G:71.5 ± 7.4	G: 73% ± 22%	-	15/28 (High)
Kozu R et al. (2011) [37]	Nagasaki Hospital	Prospective Non-RCT 4 Groups	G1: 16 (18.75) G2: 17 (23.5) G3: 17 (35.29) G4: 15 (40)	G1: 65.4 ± 7.7 G2: 67.8 ± 7.4 G3: 68.1 ± 7.6 G4: 68.7 ± 7.5	G1: 2.2 ± 0.6 G2: 1.9 ± 0.6 G3: 1.8 ± 0.6 G4:1.5 ± 0.5	-	16/28 (Low)
Kozu R et al. (2011) [38]	Nagasaki Hospital	Prospective Non-RCT 2 Groups	G1: 45 (17.7) G2: 45 (15.5)	G1: 67.5 ± 7.8 G2: 67.3 ± 5.1	G1: 2.0 ± 0.6 G2: 2.5 ± 0.7	6 months	16/28 (Low)
Ozalevli S at al. (2009) [39]	Turkey N.R.	Prospective Non-RCT 1 Group	G: 15 (33.33)	G: 62.8 ± 8.5	G:2.3 ± 0.8	-	14/28 (High)

SD: standard deviation; FVC: forced vital capacity; Dl_co_: diffusing capacity of the lungs for carbon monoxide; RCT: randomized controlled trial.

**Table 2 healthcare-11-02925-t002:** Characteristics of the included interventions.

Study (Year)	Experimental Intervention	Control	Programs (Weeks)	Frequency (Days/Week)	Dose (Total Minutes)	Supervision	Measured Outcomes	Main Findings
Choi HE et al. (2023) [30]	- Breathing techniques (chest expansion, diaphragmatic, segmental, cough, threshold) - Conventional chest physiotherapy - Aerobic training - Endurance Training	One exercise training season Recommendation	8	3	57	YES	Primary: - Exercise capacity (6 MWT, VO_2max_) Secondary: - Quality of life (SGRQ-I) - Muscle strength (handgrip strength) - Skeletal muscle mass (SMM) - Spirometry (FVC, FEV1) - Pulmonary function (Dl_co_) - Peak cough flow (PCF)	6 MWD Pre < Post (*p* = 0.013) EG > CG (NS) SGRQ-I Pre < Post (NS) EG > CG (NS) DLco Pre < Post (NS) EG > CG (NS)
Zhou M et al. (2021) [31]	- Breathing techniques - Mindfulness - Physical movements	Recommendation to maintain usual activities	4	5	35	YES	Primary: - Exercise capacity (6MWT) - Quality of life (SGRQ-I); Secondary: - Functional status (mMRC) - Spirometry (FVC, FEV1) - Pulmonary function (Dl_co_)	6MWD Pre < Post (*p* < 0.05) EG > CG (0.013) 2 m (*p* = 0.001) 6 m (*p* = 0.001) SGRQ-I Pre < Post (*p* < 0.05) EG > CG (NS) 2 m (*p* = 0.005) 6 m (NS) Dl_co_ Pre < Post (*p* < 0.05) EG > CG (NR) 2 m (NS) 6 m (NR)
Shen L et al. (2021) [32]	- Breathing techniques (deep breathing)	Usual care; follow-up	48	7	15	MIXED	Primary: - Spirometry (FVC) - Lung volume - (X-ray) Secondary: - Quality of life (SGRQ); - Spirometry (FEV1) - Exercise capacity (6MWT) - Pulmonary function (Dl_co_)	6MWD Pre < Post (*p* < 0.05) EG > CG (NS) 12 m (*p* = 0.041) SGRQ Pre < Post (*p* < 0.05) EG > CG (*p* = 0.016) 12 m (*p* = 0.003) Dl_co_ Pre < Post (*p* < 0.05) EG > CG (*p* < 0.001) 12 m (*p* = 0.003)
Jarosch I et al. (2020) [33]	- Breathing techniques - Medical Care (oxygen therapy + non-invasive ventilation) - Aerobic training - Endurance Training - Psychological support - Education	Usual care	3	5–6	NR	YES	Primary: - Exercise capacity (6MWD) Secondary: - Anxiety and depression (HADS) - Quality of life (CRQ; SF-36) - Pulmonary function (Dl_co_) - Activity levels (SenseWear Armband^®^)	6MWD Pre < Post (*p* < 0.001) EG > CG (*p* = 0.006) 3 m (NS) SF-36 Pre < Post (*p* < 0.05) Physical: EG > CG (NS) 3 m (NS) Mental: Pre < Post (*p* < 0.05) EG >CG (*p* = 0.008) 3 m (NS) Dl_co_ NR
Vainshelboim B et al. (2014) [34]	- Breathing techniques (deep breathing) - Aerobic training - Endurance training - Flexibility - Education	Usual care	12	NR	60	YES	Primary: - Exercise capacity (6MWD) Secondary: - Exercise capacity (30-SCST) - Quality of life (SGRQ) - Dyspnea (BDI) - Spirometry (FVC, FEV1, FEV1/FVC) - Cardiopulmonary function (CPET) - Functional status (mMRC) - Pulmonary function (Dl_co,_ MVV)	6MWD Pre < Post (*p* < 0.001) EG > CG (*p* < 0.001) SGRQ Pre < Post (*p* < 0.05) EG > CG (*p* < 0.001) Dl_co_ Pre < Post (*p* < 0.05) EG > CG (NS)
Rifaat N et al. (2014) [35]	- Breathing techniques (pursed-lipped, diaphragmatic breathing) - Conventional chest physiotherapy	-	8	3	NR	YES	- Exercise capacity (6MWT) - Quality of life (SGRQ) - Pulmonary function (Dl_co_) - Blood gas quality control (PaCO2; PaO2; SaO2) - Dyspnea (MBS) spirometry (FVC, % predicted; FVC, FEV1, FEV1/FVC)	6MWD Pre < Post (*p* = 0.001) SGRQ Pre < Post (*p* = 0.001) Dl_co_ Pre < Post (NS)
Swigris J et al. (2011) [36]	- Breathing techniques (pursed-lipped, diaphragmatic breathing) - Aerobic training - Endurance training - Education	-	6–8	3	NR	YES	- Quality of life (SF-36) - Exercise capacity (6MWT) - Spirometry (FVC) - Pulmonary function (Dl_co_) - Fatigue (FSS) - Anxiety (GAD) - Depression (PHQ) - Sleep quality (PSQ)	6MWD Pre < Post (*p* = 0.01) SF-36 Pre < Post (NS) Dl_co_ Pre < Post (NS)
Kozu R et al. (2011) [37]	- Breathing techniques (control techniques) - Endurance training - Relaxation - Education	-	8	4–5	90	YES	Primary: - Quality of life (SF-36) - Exercise capacity (6MWD) - SaO2 Secondary: - Dyspnea (Borg) - Activities of daily living - Heart rate (Polar A1) - Muscle strength (quadriceps and handheld dynamometer) - Pulmonary function (Dl_co_)	6MWD Pre < Post (*p* < 0.05) SF-36 Pre < Post (*p* < 0.05) Dl_co_ Pre < Post (*p* < 0.05)
Kozu R et al. (2011) [38]	- Breathing techniques (control techniques, pursed-lipped breathing) - Stretching - Strength and Endurance training - Relaxation - Education	A COPD cohort	8	2	90	YES	- Quality of life (Sf-36) - Exercise capacity (6MWT) - Pulmonary function (Dl_co_) - Activities of daily living - Blood gas quality control (PaO2: PaCO2) - Dyspnea (BDI; TDI) - Functional status (MRC) - Muscle strength (quadriceps and handheld dynamometer)	6MWD Pre < Post (*p* < 0.001) 6 m (NS) EG < CG (*p* < 0.001) 6 m (*p* < 0.001) SF-36 Pre < Post (NS) 6 m (*p* < 0.05) EG < CG (*p* = 0.0012) 6 m (*p* = 0.035) Dl_co_ Pre < Post (NR) 6 m (*p* < 0.001) EG < CG (NR)
Ozalevli S at al. (2009) [39]	- Breathing techniques (pursed-lipped breathing, thoracic expansions, diaphragmatic breathing, control techniques) - Aerobic training	-	12	5	15–30	NO	- Quality of life (SF-36) - Exercise capacity (6MWT) - Dyspnea (MBS) - Spirometry (FVC, FEV1, FEV1/FVC) - Functional status (MRCS) - Pulmonary function (PFT, Dl_co_)	6MWD Pre < Post (*p* = 0.04) SF-36 Pre < Post (*p* = 0.04) Dl_co_ Pre < Post (NR)

EG: experimental group; CG: control group; NR: non-reported; SGRQ: behavior of quality of life; 6MWT: 6-Minute Walking Test; mMRC: modify Medical Research Council; FVC: forced vital capacity; FEV-1: forced expiratory volume-1 s; Dl_co_: diffusing capacity of the lungs for carbon monoxide; RRPF: respiratory rehabilitation for pulmonary fibrosis; HADs: Hospitality Anxiety and Depression Score; CRQ: Chronic Respiratory Disease Questionnaire; SF-36: Short Form-36 survey; ILET: Incremental Load Ergometry Test; CLET: Constant Load Ergometry Test; ISWT: Incremental Shuttle Walk Test; BDI: Baseline Dyspnea Index; MIP: maximum inspiratory pressure; MEP: maximum expiratory pressure; 30-SCST: 30 s chair stand; CPET: cardiopulmonary exercise testing; MVV: maximum voluntary ventilation; PaCO2: carbone dioxygen partial pressure; PaO2: oxygen partial pressure; SaO2: oxygen saturation; MBS: Modified Borg Scale; GAD: General Anxiety Disorder-7; PHQ: Patient Health Questionnaire-8; PSQ: Pittsburgh Sleep Quality Index; FSS: Fatigue Severity Scale; TDI: Transition Dyspnea Index; PFT: Pulmonary Function Test.

## Data Availability

Not additional data are available.

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
