# Peer review of "Effect of Chest Physiotherapy on Quality of Life, Exercise Capacity and Pulmonary Function in Patients with Idiopathic Pulmonary Fibrosis: A Systematic Review and Meta-Analysis"

_healthcare, 2023, doi:10.3390/healthcare11222925_

Round 1

Reviewer 1 Report

Comments and Suggestions for Authors

The authors aimed to investigate the effect that a respiratory training rehabilitation could have on QOL, exercise capacity and pulmonary function in people with idiopathic pulmonary fibrosis. A pulmonary disease with signifant impact on patients life. Although, pharmacological treatments are developing, pulmonary rehabilitation is a key component in managing the symptoms of the disease and sustaining functionality.

Major comments:

1. Although the authors performed this systematic review in accordance to cochrane guidlines, there is a major consern regarding the nature of the intervention that was investigated. The authors used the term respiratory training but were not described it fully and providing the appropriate reference. It seems that the authors talk about pulmonary rehabilitation, yet they avoid naming it so. Even in the picos' table they mention the use of respiratory training along with othe  rehabilitation strategies, but still without describing what this is. This create a questionable heterogenity that could raise serious question on the meta- analysis performed. The authors need to clearly describe what the examined intervention includes - is it pulmonary rehabilitation or respiratory muscle trraining ? 

2. This evokes problems in the section of the discussion which is poorly presented. There is little interpretention of the findings.

3. Risk of bias is not clearly presented, besides the quality of the studies

Minor comments:

1.The authors fail to mention that there are quite a lot systematic reviews on the matter, thus providing a purpose for this one. There is no mention to limitations or clinical considerations

2. The results in table 2 should present the main significant findings in relation to the outcomes investigated by this systemativ review. Fewer words but more numbers, especially p value.

Author Response

Healthcare Journal

Ref.: healthcare-2669889
Effect of respiratory training rehabilitation on quality of life, exercise capacity and pulmonary function in patients with idiopathic pulmonary fibrosis: A systematic review and meta-analysis.

Dear Editor and Reviewers,

Please find a revision of our manuscript entitled "Effect of respiratory training rehabilitation on quality of life, exercise capacity and pulmonary function in patients with idiopathic pulmonary fibrosis: A systematic review and meta-analysis." We would like to thank the reviewers for the comments. Changes have been highlighted in yellow in the revised manuscript. An itemized point-by-point response to comments is presented below.

Reviewer #1: Special comments

Major comments:

Comment 1: Although the authors performed this systematic review in accordance to cochrane guidlines, there is a major consern regarding the nature of the intervention that was investigated. The authors used the term respiratory training but were not described it fully and providing the appropriate reference. It seems that the authors talk about pulmonary rehabilitation, yet they avoid naming it so. Even in the picos' table they mention the use of respiratory training along with othe  rehabilitation strategies, but still without describing what this is. This create a questionable heterogenity that could raise serious question on the meta- analysis performed. The authors need to clearly describe what the examined intervention includes - is it pulmonary rehabilitation or respiratory muscle training ?

Response: We acknowledge the reviewer’s comment. We have reformulated the terminology of the intervention. We have repeated the search and have rewritten entire sections to make it more adequate.

Comment 2: This evokes problems in the section of the discussion which is poorly presented. There is little interpretention of the findings.

Response: We acknowledge the reviewer’s comment and have rewritten the section to improve it.

Comment 3:  Risk of bias is not clearly presented, besides the quality of the studies

Response: We acknowledge the reviewer’s comment and have added a section to improve the manuscript.

Minor comments:

Comment 4:  The authors fail to mention that there are quite a lot systematic reviews on the matter, thus providing a purpose for this one. There is no mention to limitations or clinical considerations.

Response: We acknowledge the reviewer’s comment and have added two sections to make things clearer.

Comment 5:  The results in table 2 should present the main significant findings in relation to the outcomes investigated by this systematic review. Fewer words but more numbers, especially p value.

Response: We acknowledge the reviewer’s comment and we have rewritten the results of Table 2 to improve the manuscript.

Reviewer 2 Report

Comments and Suggestions for Authors

Dear authors,

First and foremost, I would like to commend you on your work regarding the "Effect of respiratory training rehabilitation on quality of life, exercise capacity, and pulmonary function in patients with idiopathic pulmonary fibrosis: A systematic review and meta-analysis." The relevance and importance of the topic addressed is undeniable, and it holds significant potential for advancing our understanding and potential interventions for idiopathic pulmonary fibrosis.

However, upon careful review, I have identified a series of concerns and errors within the manuscript that raise questions about the meticulousness of its execution. While the overarching theme and direction of the research are commendable, these discrepancies, if unaddressed, may detract from the paper's credibility and impact.

I kindly suggest a thorough review and rectification of these issues to uphold the rigor and integrity that a study of this magnitude and importance deserves.

1 Comment:

Your manuscript requires expert English editing.

2 Comment in reference to the Abstract

2.1 For a meta-analysis, it's crucial to provide specific effect measures or statistics in the results, such as effect size, confidence intervals, or p-values. Please incorporate these.

2.2 The objective should align with the nature of the study being conducted. The current objective resembles that of a clinical trial rather than a systematic review and meta-analysis. Please revise to ensure consistency with your study design.

2.3 In the methodology section of the abstract: While you mention that a systematic review and meta-analysis were conducted, it would be helpful to clarify whether all 12 selected studies were included in the meta-analysis or just a subset of them. Please provide this specificity for clarity.

2.4 In the results section of the abstract:

Instead of "We selected 12 studies which perform a respiratory training program," a more appropriate phrasing would be "We included 12 studies that evaluated respiratory training programs."

You've noted "significative heterogeneity" in the conducted studies. Incorporating a specific measure of this heterogeneity, such as the I^2 value, would give readers a clearer, quantitative understanding of the extent of heterogeneity present.

2.5 In the conclusion section of the abstract: The phrase "beneficial effects for reducing the presented disability" is somewhat ambiguous. It would be more informative to specify the main findings or benefits observed in terms of disability reduction.

3 Comment in reference to the Introduction

3.1 The initial description of Idiopathic Pulmonary Fibrosis (IPF) is succinct. However, it would be beneficial to provide a more detailed account of its etiology, prevalence, and related aspects. Consider starting with a broader context on pulmonary fibrosis or lung diseases and then narrowing the focus to IPF specifically.

3.2 We noticed that some sentences, particularly in lines 34-39 and 56-58, make broad claims without appropriate references. It is essential that every factual statement is substantiated with relevant citations. While the introduction utilizes sequential referencing, it is pivotal, as emphasized by PRISMA and Cochrane guidelines, to maintain consistency and specificity in citations. It's not logical to support a single sentence with five references, as observed in lines 34-39. Please review and adjust accordingly.

3.3 We also observed general grammar issues throughout the introduction. Fragmented sentences and occasional word choice errors affect the overall readability. We recommend a thorough grammar and phrasing review to enhance clarity and coherence.

- In line 33, the correct phrasing should be "...consists of a fibrotic..." instead of "...consists on a fibrotic...".

- For line 42, consider using "appearance" or "onset" in place of "apparition" to convey the intended meaning more accurately.

3.4 Scientific Terminology (Line 37): It might be helpful to explain the abbreviations and clinical tests such as "forced vital capacity" to ensure clarity for all readers.

Consider including definitions or brief descriptions for specific terms or phenomena that aren't universally understood, like "dynamic hyperinflation".

3.5 Clarity and Flow: The transition between paragraphs could be smoother. It's crucial to maintain a logical flow when discussing the disease's progression, presentation, complications, and treatments.

3.6 Rationale for the Review: While the need for a review on the topic is mentioned, it could be emphasized more. Why now? What's lacking in the current literature?

3.7 Objective Statement: The objective (lines 58-60) is clear but could be highlighted or separated for emphasis.

Consider restructuring the statement to start with the aim: "The aim of this study is to... due to... and...".

3.8 Intervention Clarification: The introduction lists various respiratory training techniques (line 55). It may benefit from clarity on whether the meta-analysis will evaluate all these techniques collectively or individually.

3.9 Relevance of the Study: Briefly discuss the potential implications of this study for clinical practice. How might the results influence treatment strategies or guidelines for IPF?

3.10 The introduction appears somewhat brief and lacks depth. I recommend enhancing the justification for the study by highlighting any controversies or recent developments in IPF and respiratory training. Additionally, provide a more comprehensive overview of the current state of knowledge, especially regarding respiratory training efficacy. Lastly, emphasize the clinical or research significance of the problem, illustrating its relevance to the medical and scientific community.

4. Comment in reference to the Methodology

4.1 We noticed that in your methodology, you mention that you conducted the review following the PRISMA guidelines (Preferred Reporting Items for Systematic Reviews and Meta-Analyses). To ensure transparency, replicability, and methodological integrity of your review, it would be highly beneficial if you include an appendix with the properly completed PRISMA checklist. This will allow reviewers and readers to accurately assess to what extent you adhered to these recognized guidelines. Including this will strengthen the quality and credibility of your work.

4.2 We have noticed a discrepancy between the dates reported in your manuscript and those recorded in PROSPERO. As you know, PROSPERO is a crucial database for registering systematic review protocols, and its purpose is to provide complete transparency to the review process and prevent redundant reviews on the same topic. It is essential that the information is consistent and up-to-date across both sources to ensure the integrity and validity of the process. We urge you to review and update your PROSPERO registration to align with what is presented in your manuscript.

4.3 Study Registration: While you've registered the study, the purpose and objectives of this systematic review have not been clearly outlined. What exact question is the review aiming to address?

4.4 Search Strategy: It would be beneficial to provide the exact search strings you used for each database. You mentioned testing and refining the search strategy, but there's no mention of who conducted the searches or if multiple reviewers were involved. If so, did they work independently? Were there any disagreements, and how were they resolved? Additionally, please attach a table detailing the search strategy employed, describing the process across the different databases. This addition will provide clarity and ensure a comprehensive understanding of your methodology.

4.5 Selection Criteria: For the PICOS criteria, you have provided a list, but you've not clearly defined the study designs you are including (e.g., RCTs, observational studies).

Regarding the exclusion criteria: what do you mean by "Studies without any affinity"? This is vague and needs further clarification.

4.6 Data Extraction: It's unclear who performed the data extraction. Was it done independently by multiple reviewers? Were there any disagreements during data extraction, and how were they resolved?

4.7 Quality Assessment: The criteria for categorizing studies as excellent, good, fair, or poor based on the Downs and Black quality assessment method is mentioned. Still, it's unclear how discrepancies or disagreements among reviewers, if any, were handled.

4.8 Risk of Bias: Were all the seven domains from the Cochrane Collaboration Risk of Bias Tool assessed for each study? It's essential to provide more detailed information on the domains evaluated and the results.

4.9 Meta-analysis: It would be beneficial to clarify how missing data (if any) were handled in the analysis, beyond contacting the authors.

You've mentioned using both random and fixed effects models based on the I^2 statistic; however, a threshold for significant heterogeneity is not provided.

It's important to specify the criteria for excluding trials from sensitivity analysis due to high risk.

4.10 The methodology seems to lack information on how potential conflicts of interest were managed among the reviewers.

There's no mention of the process of resolving disagreements during study selection, data extraction, and quality assessment.

5. Comment in reference to the Results

5.1 In alignment with PRISMA recommendations, please structure your results section with distinct subsections, mirroring the detail of your methodology. Specifically, provide clear divisions and details on: study selection, study characteristics, individual study risk of bias, individual study results, synthesis outcomes, publication biases, and evidence certainty. This refined structure will enhance clarity and allow for a comprehensive understanding of your findings.

5.2 Upon reviewing your manuscript, we noticed that while you reference the Cochrane guidelines for risk of bias, the presentation of this assessment in your results could be more comprehensive. It's imperative to include graphical representations of the risk of bias for each individual study analyzed, as per Cochrane's recommendations. These graphs provide a visual summary and lend more clarity to the assessment, facilitating understanding for both reviewers and readers. By incorporating this, you will be enhancing the transparency, depth, and comprehensiveness of your risk of bias evaluation, aligning more closely with best practices.

We suggest making these additions to ensure that your work aligns with the rigorous standards set by the Cochrane Collaboration, thereby bolstering the credibility and robustness of your findings.

5.3 When conducting a systematic review, particularly with the goal of assessing the efficacy of an intervention, in this case, respiratory training in patients with Idiopathic Pulmonary Fibrosis (IPF), it is vital that the results tables highlight pertinent statistical information. It's imperative to include at the very least the primary and secondary outcome measures, in line with the variables assessed in the studies (e.g., pulmonary function, quality of life). Moreover, the 'p' value, indicating statistical significance, should be clearly presented. We suggest enhancing Table 2 to reflect this essential information. This not only facilitates a clear and concise understanding for readers but also bolsters the transparency and credibility of your findings.

5.4 In accordance with PRISMA guidelines for systematic reviews, I recommend enhancing your tables to provide comprehensive information. In "Table 1: Description of the Studies," consider adding columns for "Location/Country," "Funding Source," and "Duration of Follow-up." For "Table 2: Description of the Interventions," it would be beneficial to include columns detailing "Measured Outcomes," "Main Findings," and their respective statistical values. These additions will improve the clarity and thoroughness of your presentation, ensuring readers have a complete understanding of the evaluated studies and interventions.

5.5 To enhance clarity and facilitate a more comprehensive understanding for the readers, I recommend separating the columns "Programs (Weeks)", "Frequency (Days/Week)", and "Dose (Total Minutes)" in your table. By providing distinct columns for each of these parameters, you allow for a more straightforward interpretation of the specific details of each intervention evaluated.

5.6 In your manuscript, while the tables provide a comprehensive view of both qualitative and quantitative analyses, it's imperative to elaborate on these findings within the text, especially concerning the meta-analysis results. A meta-analysis synthesizes a wealth of data, and its findings are of utmost importance; hence, they need to be articulated with precision in the narrative. This ensures that readers not only see the aggregated data but also comprehend its significance in the context of your review. By doing so, you offer a more nuanced understanding, bridging the gap between raw data and its overarching implications. Adhering to best practice guidelines, a detailed narrative accompanying tables and specific meta-analysis results will ensure clarity, enhance comprehensibility, and bolster the credibility of your work. We strongly recommend expanding on the results presented in your tables, particularly the meta-analysis outcomes, in the body of your manuscript for a more thorough and accessible presentation.

5.7. Upon reviewing your manuscript, we noticed an omission in Table 2 concerning the intervention details for the control group. It's essential to provide a clear description of what was provided to the control group, be it standard care, a placebo, or another form of intervention. Such information ensures that readers can fully comprehend the baseline against which the active interventions were compared. This is particularly vital in systematic reviews and meta-analyses where understanding the context and content of control interventions is crucial for the correct interpretation of the findings. We strongly recommend updating Table 2 to include a column or section dedicated to detailing the control group's intervention. This inclusion will enhance the clarity and transparency of your work, ensuring a comprehensive understanding of your study parameters and outcomes.

5.8 Upon meticulous review of your manuscript and the references you've included, it appears there may have been an oversight in your literature search or inclusion criteria. We've identified relevant studies that align with your study's objective but seem to be absent from your review. One prime example is the study by Kataoka K, et al., titled "Long-term effect of pulmonary rehabilitation in idiopathic pulmonary fibrosis: a randomised controlled trial," published in Thorax in 2023. This study provides valuable insights into the area you are investigating and seems to fit within your selected criteria.

Given the rigorous standards of systematic reviews and the importance of comprehensiveness in capturing all pertinent literature, we recommend performing a more thorough and updated search. It would be beneficial to incorporate any relevant studies that may have been inadvertently omitted to ensure the completeness and robustness of your findings. Including such studies will not only enhance the validity and reliability of your review but also strengthen its overall contribution to the field. Please consider revisiting your literature search strategy and integrating these essential references into your work.

6 Comment in reference to the Discussion and Conclusions

6.1 Repetitiveness. The effects of respiratory training programs on IPF patients' pulmonary capacity, gas diffusion, and quality of life are repeatedly emphasized. A concise presentation is critical for clarity. Instead of reiterating the benefits multiple times, the discussion should weave these results into the broader landscape of IPF treatment.

6.2 Reference Context. The discussion cites numerous references, but there's a lack of detail about these studies' specific findings. For instance:

When mentioning global entities like the WHO and the American Centers for Disease Control and Prevention, what were their specific recommendations about respiratory training for IPF patients?

What did the cited studies ([26,40-42]) specifically discover about the effects of supervised exercise on pulmonary fibrosis?

6.3 Comparison with Other Reviews. It's mentioned that previous reviews have found high efficacy in pulmonary rehabilitation. What were the distinguishing factors in this review? Were the methodologies or inclusion criteria different? Highlighting these distinctions would give the reader a better understanding of this review's unique contributions.

6.4 Psycho-emotional Aspects. The improvements in psycho-emotional aspects are briefly discussed. This can be elaborated on:

What specific psycho-emotional outcomes were measured in the studies?

How significant were these improvements, and how do they compare with other interventions for IPF?

6.5 Limitations:

Heterogeneity of Study Design. The limitation section acknowledges the heterogeneity in the study designs. This needs further unpacking:

Were there any specific study designs that were overrepresented?

How might these designs introduce biases or affect the overall findings?

6.6 Multidimensionality of Interventions. The complexity of intervention programs is briefly mentioned. Delving deeper:

Were there common components across all interventions?

Which components are believed to have the most therapeutic potential?

How does this multidimensionality affect the reproducibility of the results in real-world settings?

6.7 Missed Studies. The potential of missing relevant studies is noted. But more context is required:

Were there date or language restrictions during the search?

Was there a regional bias in the studies included?

Given the missed study (Kataoka K, et al.), what was the reason for its omission?

6.8 Lack of Strengths Discussion. While the authors appropriately highlight some limitations of their study, they neglect to discuss the strengths of their work, especially in comparison to existing studies. Addressing strengths provides:

Balanced Insight: Presenting both strengths and limitations offers a comprehensive, balanced view of the research. It's crucial to understand what makes this review unique or valuable within the larger research context.

Guidance for Future Research: Discussing strengths can help future researchers understand what methodologies or approaches were particularly effective and could be employed in subsequent studies.

Given this oversight, the authors should:

Detail Their Unique Strengths: What methodologies, data sources, or analytical techniques did they use that might be distinct from prior studies? How does this add to the robustness or comprehensiveness of their findings?

Comparison with Existing Studies: Explicitly draw out contrasts between their methods and findings and those of prior studies, emphasizing where their work adds novel insights or clarifications.

6.9 Conclusion:

Heterogeneity Emphasis. The heterogeneity of the study design is reiterated. However, it's essential to specify how this heterogeneity affects the robustness of the review's findings.

Strength of Evidence. The phrasing "results should be taken with caution" is somewhat ambiguous. The authors should directly address the strength of their evidence, perhaps leveraging GRADE or another system to classify their confidence in the results.

Clinical Recommendations. While the review suggests incorporating respiratory training programs in IPF clinical approach, it would be beneficial to indicate:

Are there any specific types of respiratory training programs that stand out?

What patient populations would most benefit from these programs?

6.10 Conclusion Specificity. The conclusions section should tie directly back to the individual variables or outcomes the study set out to investigate. Generic or broad statements might not provide the reader with a clear understanding of the study's specific findings. Therefore:

Tailored Conclusions: For each primary and secondary outcome or variable analyzed, the authors should provide a clear, concise conclusion. This ensures clarity and avoids ambiguity about the implications of the research.

Avoid Over-generalization: Conclusions should stay within the bounds of the data and analysis presented in the study. Over-reaching or drawing conclusions not directly supported by the data can mislead readers and detract from the study's credibility.

By refining their discussion of strengths, ensuring conclusions directly address their research variables, and acknowledging and comparing their work with prior studies, the authors can significantly enhance the clarity, credibility, and impact of their review.

7 Comment in reference to the References

7.1 It's essential that your bibliographic references adhere to the specific guidelines of the journal you plan to submit to. Please review and format your citations and references according to the journal's style guide to ensure consistency and compliance with their editorial requirements. Doing so will streamline the review and publication process for your manuscript.

Comments on the Quality of English Language

Upon a thorough review of your manuscript titled "Effect of respiratory training rehabilitation on quality of life, exercise capacity and pulmonary function in patients with idiopathic pulmonary fibrosis: A systematic review and meta-analysis", I have identified several areas that require additional attention, particularly concerning grammar and clarity of the text.

 To ensure the utmost quality of your work, it is essential that the manuscript be reviewed by an expert in English text editing. Below are some examples of areas that need corrections:

In the "Study registration" section, the phrase "This systematic review protocol was registered on the International Prospective Register of Systematic Reviews (PROSPERO)..." could benefit from clearer phrasing.

Under "Search strategy", the sentence "We systematically searched the articles indexed on PubMed..." displays redundancy by using "systematically" and "systematic review" in the same sentence.

Within "Meta-analysis", the segment "The Review Manager 5 (RevMan 5) software was used to realize the quantitative synthesis..." employs "realize" in a context that might be better expressed by "conduct" or "perform".

These are just a few examples, and similar issues have been observed in other parts of the text.

I urge you to undertake a comprehensive review to correct these and other grammatical and phrasing errors throughout the manuscript, thus ensuring the clarity and professionalism that a work of this significance deserves.

Author Response

Healthcare Journal

Ref.: healthcare-2669889
Effect of respiratory training rehabilitation on quality of life, exercise capacity and pulmonary function in patients with idiopathic pulmonary fibrosis: A systematic review and meta-analysis.

Dear Editor and Reviewers,

Please find a revision of our manuscript entitled "Effect of respiratory training rehabilitation on quality of life, exercise capacity and pulmonary function in patients with idiopathic pulmonary fibrosis: A systematic review and meta-analysis." We would like to thank the reviewers for the comments. Changes have been highlighted in yellow in the revised manuscript. An itemized point-by-point response to comments is presented below.

Reviewer #2: Special comments

1 Comment: Your manuscript requires expert English editing.

Response: We acknowledge the reviewer’s comment. The manuscript has now been reviewed and corrected by an expert English editor.

2 Comment in reference to the Abstract

2.1 For a meta-analysis, it's crucial to provide specific effect measures or statistics in the results, such as effect size, confidence intervals, or p-values. Please incorporate these.

2.2 The objective should align with the nature of the study being conducted. The current objective resembles that of a clinical trial rather than a systematic review and meta-analysis. Please revise to ensure consistency with your study design.

2.3 In the methodology section of the abstract: While you mention that a systematic review and meta-analysis were conducted, it would be helpful to clarify whether all 12 selected studies were included in the meta-analysis or just a subset of them. Please provide this specificity for clarity.

2.4 In the results section of the abstract:

Instead of "We selected 12 studies which perform a respiratory training program," a more appropriate phrasing would be "We included 12 studies that evaluated respiratory training programs."

You've noted "significative heterogeneity" in the conducted studies. Incorporating a specific measure of this heterogeneity, such as the I^2 value, would give readers a clearer, quantitative understanding of the extent of heterogeneity present.

2.5 In the conclusion section of the abstract: The phrase "beneficial effects for reducing the presented disability" is somewhat ambiguous. It would be more informative to specify the main findings or benefits observed in terms of disability reduction.

Response:  We acknowledge the reviewer’s comments and have rewritten the entire section to take them into account.

3 Comment in reference to the Introduction

3.1 The initial description of Idiopathic Pulmonary Fibrosis (IPF) is succinct. However, it would be beneficial to provide a more detailed account of its etiology, prevalence, and related aspects. Consider starting with a broader context on pulmonary fibrosis or lung diseases and then narrowing the focus to IPF specifically.

3.2 We noticed that some sentences, particularly in lines 34-39 and 56-58, make broad claims without appropriate references. It is essential that every factual statement is substantiated with relevant citations. While the introduction utilizes sequential referencing, it is pivotal, as emphasized by PRISMA and Cochrane guidelines, to maintain consistency and specificity in citations. It's not logical to support a single sentence with five references, as observed in lines 34-39. Please review and adjust accordingly.

3.3 We also observed general grammar issues throughout the introduction. Fragmented sentences and occasional word choice errors affect the overall readability. We recommend a thorough grammar and phrasing review to enhance clarity and coherence.- In line 33, the correct phrasing should be "...consists of a fibrotic..." instead of "...consists on a fibrotic...".- For line 42, consider using "appearance" or "onset" in place of "apparition" to convey the intended meaning more accurately.

3.4 Scientific Terminology (Line 37): It might be helpful to explain the abbreviations and clinical tests such as "forced vital capacity" to ensure clarity for all readers.

Consider including definitions or brief descriptions for specific terms or phenomena that aren't universally understood, like "dynamic hyperinflation".

3.5 Clarity and Flow: The transition between paragraphs could be smoother. It's crucial to maintain a logical flow when discussing the disease's progression, presentation, complications, and treatments.

3.6 Rationale for the Review: While the need for a review on the topic is mentioned, it could be emphasized more. Why now? What's lacking in the current literature?

3.7 Objective Statement: The objective (lines 58-60) is clear but could be highlighted or separated for emphasis. Consider restructuring the statement to start with the aim: "The aim of this study is to... due to... and...".

3.8 Intervention Clarification: The introduction lists various respiratory training techniques (line 55). It may benefit from clarity on whether the meta-analysis will evaluate all these techniques collectively or individually.

3.9 Relevance of the Study: Briefly discuss the potential implications of this study for clinical practice. How might the results influence treatment strategies or guidelines for IPF?   

3.10 The introduction appears somewhat brief and lacks depth. I recommend enhancing the justification for the study by highlighting any controversies or recent developments in IPF and respiratory training. Additionally, provide a more comprehensive overview of the current state of knowledge, especially regarding respiratory training efficacy. Lastly, emphasize the clinical or research significance of the problem, illustrating its relevance to the medical and scientific community.

Response: We have changed the entire section according to the comments received, expanding the epidemiological information, refining the references used, clarifying the abbreviations, and highlighting the importance of our review as well as the aim of the review.

  1. Comment in reference to theMethodology

4.1 We noticed that in your methodology, you mention that you conducted the review following the PRISMA guidelines (Preferred Reporting Items for Systematic Reviews and Meta-Analyses). To ensure transparency, replicability, and methodological integrity of your review, it would be highly beneficial if you include an appendix with the properly completed PRISMA checklist. This will allow reviewers and readers to accurately assess to what extent you adhered to these recognized guidelines. Including this will strengthen the quality and credibility of your work.

Response: We have added the PRISMA checklist to the files presented.

4.2 We have noticed a discrepancy between the dates reported in your manuscript and those recorded in PROSPERO. As you know, PROSPERO is a crucial database for registering systematic review protocols, and its purpose is to provide complete transparency to the review process and prevent redundant reviews on the same topic. It is essential that the information is consistent and up-to-date across both sources to ensure the integrity and validity of the process. We urge you to review and update your PROSPERO registration to align with what is presented in your manuscript.

Response: We acknowledge the reviewer’s comments and have reviewed and updated our PROSPERO registration to align with what is presented in the manuscript.

4.3 Study Registration: While you've registered the study, the purpose and objectives of this systematic review have not been clearly outlined. What exact question is the review aiming to address?

Response: The lack of a consensus about the chest physiotherapy programs that could be applied in patients with IPF, and which are the effects of these interventions on the quality of life, exercise capacity and pulmonary function of these patients.

4.4 Search Strategy: It would be beneficial to provide the exact search strings you used for each database. You mentioned testing and refining the search strategy, but there's no mention of who conducted the searches or if multiple reviewers were involved. If so, did they work independently? Were there any disagreements, and how were they resolved? Additionally, please attach a table detailing the search strategy employed, describing the process across the different databases. This addition will provide clarity and ensure a comprehensive understanding of your methodology.

Response: We acknowledge the reviewer’s comments and have added the research equations in the supplementary data.

4.5 Selection Criteria: For the PICOS criteria, you have provided a list, but you've not clearly defined the study designs you are including (e.g., RCTs, observational studies). Regarding the exclusion criteria: what do you mean by "Studies without any affinity"? This is vague and needs further clarification.

Response: We acknowledge the reviewer’s comments and have rewritten the manuscript to clarify these sections.

4.6 Data Extraction: It's unclear who performed the data extraction. Was it done independently by multiple reviewers? Were there any disagreements during data extraction, and how were they resolved?

Response: We acknowledge the comment and have added this information in the Methods section.

4.7 Quality Assessment: The criteria for categorizing studies as excellent, good, fair, or poor based on the Downs and Black quality assessment method is mentioned. Still, it's unclear how discrepancies or disagreements among reviewers, if any, were handled.

Response: We acknowledge the comment and have added this information in the Methods section.

4.8 Risk of Bias: Were all the seven domains from the Cochrane Collaboration Risk of Bias Tool assessed for each study? It's essential to provide more detailed information on the domains evaluated and the results.

Response: We have added the RoB figures to respond to this question.

4.9 Meta-analysis: It would be beneficial to clarify how missing data (if any) were handled in the analysis, beyond contacting the authors.  You've mentioned using both random and fixed effects models based on the I^2 statistic; however, a threshold for significant heterogeneity is not provided. It's important to specify the criteria for excluding trials from sensitivity analysis due to high risk.

Response: We acknowledge the comment and have added this information in the Methods section.

4.10 The methodology seems to lack information on how potential conflicts of interest were managed among the reviewers. There's no mention of the process of resolving disagreements during study selection, data extraction, and quality assessment.

Response: We acknowledge the comment and have added this information in the Methods section.

  1. Comment in reference to theResults

5.1 In alignment with PRISMA recommendations, please structure your results section with distinct subsections, mirroring the detail of your methodology. Specifically, provide clear divisions and details on: study selection, study characteristics, individual study risk of bias, individual study results, synthesis outcomes, publication biases, and evidence certainty. This refined structure will enhance clarity and allow for a comprehensive understanding of your findings.

Response: Edited.

5.2 Upon reviewing your manuscript, we noticed that while you reference the Cochrane guidelines for risk of bias, the presentation of this assessment in your results could be more comprehensive. It's imperative to include graphical representations of the risk of bias for each individual study analyzed, as per Cochrane's recommendations. These graphs provide a visual summary and lend more clarity to the assessment, facilitating understanding for both reviewers and readers. By incorporating this, you will be enhancing the transparency, depth, and comprehensiveness of your risk of bias evaluation, aligning more closely with best practices. We suggest making these additions to ensure that your work aligns with the rigorous standards set by the Cochrane Collaboration, thereby bolstering the credibility and robustness of your findings.

Response: We have added the RoB figures to respond to this question in the supplementary data.

5.3 When conducting a systematic review, particularly with the goal of assessing the efficacy of an intervention, in this case, respiratory training in patients with Idiopathic Pulmonary Fibrosis (IPF), it is vital that the results tables highlight pertinent statistical information. It's imperative to include at the very least the primary and secondary outcome measures, in line with the variables assessed in the studies (e.g., pulmonary function, quality of life). Moreover, the 'p' value, indicating statistical significance, should be clearly presented. We suggest enhancing Table 2 to reflect this essential information. This not only facilitates a clear and concise understanding for readers but also bolsters the transparency and credibility of your findings.

Response: We acknowledge the reviewer’s comments, we have adapted the table 2 in order to clarify these sections.

5.4 In accordance with PRISMA guidelines for systematic reviews, I recommend enhancing your tables to provide comprehensive information. In "Table 1: Description of the Studies," consider adding columns for "Location/Country," "Funding Source," and "Duration of Follow-up." For "Table 2: Description of the Interventions," it would be beneficial to include columns detailing "Measured Outcomes," "Main Findings," and their respective statistical values. These additions will improve the clarity and thoroughness of your presentation, ensuring readers have a complete understanding of the evaluated studies and interventions.

Response: We acknowledge the reviewer’s comment and have rewritten both tables to improve them following the reviewer’s comment.

5.5 To enhance clarity and facilitate a more comprehensive understanding for the readers, I recommend separating the columns "Programs (Weeks)", "Frequency (Days/Week)", and "Dose (Total Minutes)" in your table. By providing distinct columns for each of these parameters, you allow for a more straightforward interpretation of the specific details of each intervention evaluated.

Response: Edited.

5.6 In your manuscript, while the tables provide a comprehensive view of both qualitative and quantitative analyses, it's imperative to elaborate on these findings within the text, especially concerning the meta-analysis results. A meta-analysis synthesizes a wealth of data, and its findings are of utmost importance; hence, they need to be articulated with precision in the narrative. This ensures that readers not only see the aggregated data but also comprehend its significance in the context of your review. By doing so, you offer a more nuanced understanding, bridging the gap between raw data and its overarching implications. Adhering to best practice guidelines, a detailed narrative accompanying tables and specific meta-analysis results will ensure clarity, enhance comprehensibility, and bolster the credibility of your work. We strongly recommend expanding on the results presented in your tables, particularly the meta-analysis outcomes, in the body of your manuscript for a more thorough and accessible presentation.

Response: We acknowledge the reviewer’s comment and have rewritten the Results section to improve it following the reviewer’s comment.

5.7. Upon reviewing your manuscript, we noticed an omission in Table 2 concerning the intervention details for the control group. It's essential to provide a clear description of what was provided to the control group, be it standard care, a placebo, or another form of intervention. Such information ensures that readers can fully comprehend the baseline against which the active interventions were compared. This is particularly vital in systematic reviews and meta-analyses where understanding the context and content of control interventions is crucial for the correct interpretation of the findings. We strongly recommend updating Table 2 to include a column or section dedicated to detailing the control group's intervention. This inclusion will enhance the clarity and transparency of your work, ensuring a comprehensive understanding of your study parameters and outcomes.

Response: We acknowledge the comment and have added a column with the control intervention in Table 2. 

5.8 Upon meticulous review of your manuscript and the references you've included, it appears there may have been an oversight in your literature search or inclusion criteria. We've identified relevant studies that align with your study's objective but seem to be absent from your review. One prime example is the study by Kataoka K, et al., titled "Long-term effect of pulmonary rehabilitation in idiopathic pulmonary fibrosis: a randomised controlled trial," published in Thorax in 2023. This study provides valuable insights into the area you are investigating and seems to fit within your selected criteria.

Given the rigorous standards of systematic reviews and the importance of comprehensiveness in capturing all pertinent literature, we recommend performing a more thorough and updated search. It would be beneficial to incorporate any relevant studies that may have been inadvertently omitted to ensure the completeness and robustness of your findings. Including such studies will not only enhance the validity and reliability of your review but also strengthen its overall contribution to the field. Please consider revisiting your literature search strategy and integrating these essential references into your work.

Response: We appreciate the reviewer's comment. We have repeated the systematic search adding new terms given the change in terminology of the evaluated interventions. However, we have not found any new results to include in our review. We have reviewed the article you propose to include and thank you for your input. Unfortunately, this article does not meet the criteria of our PICOS strategy to be included.

6 Comment in reference to the Discussion and Conclusions

6.1 Repetitiveness. The effects of respiratory training programs on IPF patients' pulmonary capacity, gas diffusion, and quality of life are repeatedly emphasized. A concise presentation is critical for clarity. Instead of reiterating the benefits multiple times, the discussion should weave these results into the broader landscape of IPF treatment.

6.2 Reference Context. The discussion cites numerous references, but there's a lack of detail about these studies' specific findings. For instance:When mentioning global entities like the WHO and the American Centers for Disease Control and Prevention, what were their specific recommendations about respiratory training for IPF patients? What did the cited studies ([26,40-42]) specifically discover about the effects of supervised exercise on pulmonary fibrosis?

6.3 Comparison with Other Reviews. It's mentioned that previous reviews have found high efficacy in pulmonary rehabilitation. What were the distinguishing factors in this review? Were the methodologies or inclusion criteria different? Highlighting these distinctions would give the reader a better understanding of this review's unique contributions.

6.4 Psycho-emotional Aspects. The improvements in psycho-emotional aspects are briefly discussed. This can be elaborated on: What specific psycho-emotional outcomes were measured in the studies? How significant were these improvements, and how do they compare with other interventions for IPF?

6.5 Limitations:Heterogeneity of Study Design. The limitation section acknowledges the heterogeneity in the study designs. This needs further unpacking: Were there any specific study designs that were overrepresented? How might these designs introduce biases or affect the overall findings?

6.6 Multidimensionality of Interventions. The complexity of intervention programs is briefly mentioned. Delving deeper: Were there common components across all interventions? Which components are believed to have the most therapeutic potential? How does this multidimensionality affect the reproducibility of the results in real-world settings?

6.7 Missed Studies. The potential of missing relevant studies is noted. But more context is required: Were there date or language restrictions during the search? Was there a regional bias in the studies included? Given the missed study (Kataoka K, et al.), what was the reason for its omission?

6.8 Lack of Strengths Discussion. While the authors appropriately highlight some limitations of their study, they neglect to discuss the strengths of their work, especially in comparison to existing studies. Addressing strengths provides:  Balanced Insight: Presenting both strengths and limitations offers a comprehensive, balanced view of the research. It's crucial to understand what makes this review unique or valuable within the larger research context. Guidance for Future Research: Discussing strengths can help future researchers understand what methodologies or approaches were particularly effective and could be employed in subsequent studies. Given this oversight, the authors should: Detail Their Unique Strengths: What methodologies, data sources, or analytical techniques did they use that might be distinct from prior studies? How does this add to the robustness or comprehensiveness of their findings? Comparison with Existing Studies: Explicitly draw out contrasts between their methods and findings and those of prior studies, emphasizing where their work adds novel insights or clarifications.

Response: We appreciate the reviewer's comment. We have rewritten the entire Discussion section to make it more appropriate in light of the reviewer's comments. We have tried to be more concise in our expression and added a more detailed description of what is included in the articles we referenced, what the other reviews include and what the emotional aspects are. In addition, we have specified the limitations of the heterogeneity observed and the strengths of our study.

6.9 Conclusion:

Heterogeneity Emphasis. The heterogeneity of the study design is reiterated. However, it's essential to specify how this heterogeneity affects the robustness of the review's findings.

Strength of Evidence. The phrasing "results should be taken with caution" is somewhat ambiguous. The authors should directly address the strength of their evidence, perhaps leveraging GRADE or another system to classify their confidence in the results.

Clinical Recommendations. While the review suggests incorporating respiratory training programs in IPF clinical approach, it would be beneficial to indicate:

Are there any specific types of respiratory training programs that stand out?

What patient populations would most benefit from these programs?

Response: We acknowledge the comment and have specified how heterogeneity affects our results and conclusions. Additionally, we have clarified the strength of our results according to the RoB and Down and Black evaluations. We have also emphasized the clinical implications and the clinical application of chest physiotherapy.

6.10 Conclusion Specificity. The conclusions section should tie directly back to the individual variables or outcomes the study set out to investigate. Generic or broad statements might not provide the reader with a clear understanding of the study's specific findings. Therefore:

Tailored Conclusions: For each primary and secondary outcome or variable analyzed, the authors should provide a clear, concise conclusion. This ensures clarity and avoids ambiguity about the implications of the research.

Avoid Over-generalization: Conclusions should stay within the bounds of the data and analysis presented in the study. Over-reaching or drawing conclusions not directly supported by the data can mislead readers and detract from the study's credibility.

By refining their discussion of strengths, ensuring conclusions directly address their research variables, and acknowledging and comparing their work with prior studies, the authors can significantly enhance the clarity, credibility, and impact of their review.

Response: We acknowledge the reviewer’s comments and have provided a clear conclusion of our findings.

7 Comment in reference to the References

7.1 It's essential that your bibliographic references adhere to the specific guidelines of the journal you plan to submit to. Please review and format your citations and references according to the journal's style guide to ensure consistency and compliance with their editorial requirements. Doing so will streamline the review and publication process for your manuscript.

Response: We acknowledge the reviewer’s comments and have rewritten the bibliographic references according to the specific guidelines of the healthcare journal.

Comments on the Quality of English Language

Upon a thorough review of your manuscript titled "Effect of respiratory training rehabilitation on quality of life, exercise capacity and pulmonary function in patients with idiopathic pulmonary fibrosis: A systematic review and meta-analysis", I have identified several areas that require additional attention, particularly concerning grammar and clarity of the text.

 To ensure the utmost quality of your work, it is essential that the manuscript be reviewed by an expert in English text editing. Below are some examples of areas that need corrections:

In the "Study registration" section, the phrase "This systematic review protocol was registered on the International Prospective Register of Systematic Reviews (PROSPERO)..." could benefit from clearer phrasing.

Under "Search strategy", the sentence "We systematically searched the articles indexed on PubMed..." displays redundancy by using "systematically" and "systematic review" in the same sentence.

Within "Meta-analysis", the segment "The Review Manager 5 (RevMan 5) software was used to realize the quantitative synthesis..." employs "realize" in a context that might be better expressed by "conduct" or "perform".

These are just a few examples, and similar issues have been observed in other parts of the text.

I urge you to undertake a comprehensive review to correct these and other grammatical and phrasing errors throughout the manuscript, thus ensuring the clarity and professionalism that a work of this significance deserves.

Response: We appreciate the reviewer's comments and have sent the manuscript to an expert English editor for a comprehensive review to correct the grammatical and phrasing errors.

Round 2

Reviewer 1 Report

Comments and Suggestions for Authors

The authors made all appropriate changes, re- wrote the sections accordingly and provided an improved manuscript especially in terms of methodology.

Discussion could be firther enriched by adding the physiological explanation of the positive effect that chest had on the examined outcomes. Additional clinical considerations or recommendations could add value to this work, as the study presents the benefits of a significant clinical intervention for this population.

Author Response

REVIEWER1

Comment 1: The authors made all appropriate changes, re- wrote the sections accordingly and provided an improved manuscript especially in terms of methodology.

Discussion could be firther enriched by adding the physiological explanation of the positive effect that chest had on the examined outcomes. Additional clinical considerations or recommendations could add value to this work, as the study presents the benefits of a significant clinical intervention for this population.

Response: We acknowledge the appreciation of the changes we made in the manuscript by the reviewer, and we agree with the suggested clinical value to add in the discussion and conclusion section.

Reviewer 2 Report

Comments and Suggestions for Authors

Dear Authors,

Upon reviewing the revised version of your manuscript titled "Effect of chest physiotherapy on quality of life, exercise capacity, and pulmonary function in patients with idiopathic pulmonary fibrosis: A systematic review and meta-analysis," I noticed that while you have addressed some of the recommendations provided in the initial review, there are errors that were highlighted initially that remain uncorrected.

It is essential for you to provide a detailed list of corrections made, indicating the specific page and line numbers where these changes have been implemented. Doing so not only facilitates the review process but also ensures a thorough evaluation of each modification.

In addition to this, we have identified further changes that are required:

Please condense the abstract to a maximum of 250 words, ensuring it provides a concise overview without omitting essential information.

Despite our initial feedback pointing out various grammatical and language errors, we still observe some persistent issues. We strongly recommend that you obtain a thorough review by a language expert, and attach a proofreading report with your revised manuscript. This will ensure clarity, format consistency, and the elimination of grammatical errors, thus enhancing the overall quality of the submission.

Here's an example from the first sentence: "...dysfunction of the normal activity of alveolar epithelial tissues [1)]." The closing parenthesis after the reference "[1]" is unnecessary and should simply be "[1]".

Regarding the introduction:

- There are occasions where the sentence structure could be improved for clarity, as in "...dysfunction of the normal activity of alveolar epithelial tissues...".

- Some statements, like "It represents one end of a spectrum of types of tissue responses to injury," lack the necessary precision to fully understand their meaning in the presented context.

- I noticed repetition in certain information, particularly about the severity of IPF. It would be beneficial to consolidate these points to avoid redundancy.

- The section on the clinical management of IPF and the role of drug therapy might benefit from further clarification to avoid potential confusion.

- The transition between the discussion on the heterogeneity of the studies and the decision to undertake a systematic review is somewhat abrupt. A transition sentence might help improve the flow of the text.

The systematic review presented utilizes the PICOS model for defining eligibility criteria, but lacks a detailed breakdown of each component. It is recommended to provide more information about the studied population, specify the interventions considered, clarify comparison criteria, detail the outcome measures used, and clearly define the types of studies included. For transparency and replicability, it's vital that each component of PICOS is accurately described.

Protocol and registration: While it's mentioned that the review was registered with PROSPERO, it's not indicated if the systematic review protocol is publicly available for other researchers to consult.

Eligibility criteria: While eligibility criteria are provided, it may not be clear how these criteria were defined and if they were predetermined prior to the review.

Study selection process: It's mentioned that two researchers carried out the search, and discrepancies were resolved with a third researcher. However, there is no detailed explanation of how these discrepancies were exactly handled or if any specific method or tool was used for the selection process.

Data extraction: Although it's mentioned that data extraction was carried out using predefined categories, there's no specification about what these categories are and how they were designed.

Outcome measures: There should be specifics about which primary and secondary outcome measures were considered before starting the review.

10º Search in other sources: Although references of relevant reviews were checked, it's not clear if other sources, like clinical trial registers, were searched to identify unpublished or ongoing studies.

11º Quality assessment and risk of bias: While the tools used to assess quality and risk of bias are described, there's no detailed description of the process nor how the results were interpreted and used in the analysis.

12º Data synthesis: Random-effects and fixed-effect models are mentioned concerning heterogeneity, but it's not specified how it was decided which model to use in case of insignificant heterogeneity.

13º Additional meta-analysis: It's not clear if other types of analyses, like subgroup analysis or meta-analysis of individual studies, were considered.

14º Quality of evidence assessment: No specific tool or method (e.g., GRADE) is mentioned to assess the overall quality of the evidence for each significant outcome.

15º We recommend incorporating the GRADE (Grading of Recommendations, Assessment, Development, and Evaluations) approach in your systematic review process. This will not only enhance the credibility and clarity of your review but will also provide a systematic method for evaluating the overall quality of evidence for each significant outcome. Implementing GRADE can offer a more structured assessment and clear interpretation of the strengths and limitations of the evidence, thereby adding value to your conclusions and recommendations.

16º We recommend that within the results section, specifically under the 'Selected Studies' subsection, you provide a detailed explanation of the flow diagram that clearly illustrates the study selection process. This diagram should break down the total number of studies identified, the number of studies excluded with specific reasons, and the studies that were ultimately included in the systematic review and those that were meta-analyzed. It's essential that the flow diagram clearly differentiates between the 12 studies that proceeded to qualitative analysis and those selected for meta-analysis. Furthermore, for the 38 articles that reported on exclusion, it would be beneficial for the flow diagram to detail the specific reasons for exclusion for each one, thereby providing greater transparency and understanding of the selection process.

17º After a detailed review of the corrections and responses provided in relation to the comments from the first review, we have noticed that some of the recommendations and suggested changes have been overlooked or ignored, particularly in the results section.

I would like to especially highlight the article by Kataoka K, et al., titled "Long-term effect of pulmonary rehabilitation in idiopathic pulmonary fibrosis: a randomized controlled trial." Although you mention that this article does not meet your search criteria, we would like to emphasize that it seems to align perfectly with your PICOS strategy. This study is conducted in patients with idiopathic pulmonary fibrosis, the intervention is pulmonary rehabilitation, there is a control group, the variable 6MWD that you consider is analyzed, and it is a clinical trial. Given this context, could you specify the exact reason for its exclusion?

We understand that conducting systematic reviews is a complex process, but it is essential to ensure the inclusion of all relevant studies to guarantee the validity and robustness of your findings. We urge you to reconsider the inclusion of this article and to reformulate the results section accordingly.

18º Among other overlooked aspects from the first review, you failed to indicate whether the included studies received funding, provide a detailed explanation regarding the risk of bias in the selected studies, offer an in-depth description within the text of the tables presented, and give a comprehensive account of the results from each of the chosen studies. It should be noted that simply reporting doesn't suffice; a concise summary outlining intra-group and inter-group changes is necessary, among other considerations.

It's paramount that you closely review the feedback provided in the initial review and ensure strict adherence to all the specified recommendations.

19º Discussion Section:

- Clinical Relevance and Contextualization: Within your discussion, it would be beneficial to provide a deeper context at the outset that emphasizes the clinical significance of chest physiotherapy in patients with idiopathic pulmonary fibrosis. By focusing on the relevance of the intervention, readers can better grasp the potential impact of the findings on actual medical practice. Why is it crucial to study chest physiotherapy specifically for these patients, and what is its potential clinical significance?

- Exploration of Controversies: You mentioned that the results in previous literature are controversial. It would be valuable if a segment of the discussion delved deeper into these controversies. Are there methodological, demographic, or contextual factors that might explain these divergent findings? A detailed exploration of these aspects would enrich the discussion and offer a more comprehensive perspective on the current state of research in this field.

- Psycho-emotional Outcomes and Their Impact: While the study touches upon psycho-emotional improvements, I'd suggest delving deeper into the discussion regarding the relevance of these improvements. How do these improvements correlate with the patients' overall quality of life? Are there other studies supporting these findings? By contextualizing and delving deeper into these results, you can provide a more holistic view of the impact of chest physiotherapy on the overall well-being of idiopathic pulmonary fibrosis patients.

Comments on the Quality of English Language

Despite our initial feedback pointing out various grammatical and language errors, we still observe some persistent issues. We strongly recommend that you obtain a thorough review by a language expert, and attach a proofreading report with your revised manuscript. This will ensure clarity, format consistency, and the elimination of grammatical errors, thus enhancing the overall quality of the submission.

Author Response

REVIEWER 2

Dear Authors,

Comment 1: Upon reviewing the revised version of your manuscript titled "Effect of chest physiotherapy on quality of life, exercise capacity, and pulmonary function in patients with idiopathic pulmonary fibrosis: A systematic review and meta-analysis," I noticed that while you have addressed some of the recommendations provided in the initial review, there are errors that were highlighted initially that remain uncorrected.

It is essential for you to provide a detailed list of corrections made, indicating the specific page and line numbers where these changes have been implemented. Doing so not only facilitates the review process but also ensures a thorough evaluation of each modification.

Response: We are sorry for the response to reviewers provided previously but we have provided a detailed point by point response to reviewer comments. The style of the journal did not permit the addition of page lines to include the detail, but in order to make our changes easy to find in the corrected manuscript we have include pages and inserted sometimes some of the changes included in the text.

Comment 2: In addition to this, we have identified further changes that are required. Please condense the abstract to a maximum of 250 words, ensuring it provides a concise overview without omitting essential information.

Response: we have revised the abstract and rewritten it to adapt it to 249 words.

Comment 3: Despite our initial feedback pointing out various grammatical and language errors, we still observe some persistent issues. We strongly recommend that you obtain a thorough review by a language expert, and attach a proofreading report with your revised manuscript. This will ensure clarity, format consistency, and the elimination of grammatical errors, thus enhancing the overall quality of the submission.

Here's an example from the first sentence: "...dysfunction of the normal activity of alveolar epithelial tissues [1)]." The closing parenthesis after the reference "[1]" is unnecessary and should simply be "[1]".

Response: We acknowledge the comment of the reviewer and we have sended the manuscript to a professional english grammar editor. We have corrected the pointed out error, but we consider that there is not a grammatical issue and is more an editing issue. In this line we have reviewed the hole manuscript.

Comment 4: Regarding the introduction:

- There are occasions where the sentence structure could be improved for clarity, as in "...dysfunction of the normal activity of alveolar epithelial tissues...".

Response: Phrase modified for clarity, please see page 1 of the manuscript.

Comment 5: Some statements, like "It represents one end of a spectrum of types of tissue responses to injury," lack the necessary precision to fully understand their meaning in the presented context.

Response:  Phrase rewritten for clarity and precision, please see pag 1 of the manuscript.

Comment 6: I noticed repetition in certain information, particularly about the severity of IPF. It would be beneficial to consolidate these points to avoid redundancy.

Response: We have revised the provided information and eliminated certain information to improve the clarity of the text.

Comment 7: The section on the clinical management of IPF and the role of drug therapy might benefit from further clarification to avoid potential confusion.

Response: we have added some clarifying information about drug therapy to improve the section, please see pag 2.

Comment 8: The transition between the discussion on the heterogeneity of the studies and the decision to undertake a systematic review is somewhat abrupt. A transition sentence might help improve the flow of the text.

Response: we agree with the reviewer about the importance to make the introduction more easy to read. We have rewritten this section, please see pag 2 of the manuscript.

Comment 9: The systematic review presented utilizes the PICOS model for defining eligibility criteria, but lacks a detailed breakdown of each component. It is recommended to provide more information about the studied population, specify the interventions considered, clarify comparison criteria, detail the outcome measures used, and clearly define the types of studies included. For transparency and replicability, it's vital that each component of PICOS is accurately described.

Response: as suggested we have expanded the description of each component of PICOS strategy. Please see pag 3:

“(1) Patients had to be adults with  a diagnosis of idiopathic pulmonary fibrosis in accordance with the clinical guidelines of the American Thoracic Society (ATS) and the European Respiratory Society (ERS);

(2) interventions had to include chest physiotherapy as described by Warnok et cols (i.e., conventional chest physiotherapy, positive expiratory pressure mask therapy, high pressure PEP mask therapy, active cycle of breathing techniques, autogenic drainage, exercise and oscillating devices) [21] isolated or in combination with other techniques.

(3) the comparator group was to be standard medical care or programs without chest physiotherapy;

(4) included outcomes are respiratory function, exercise capacity and/or quality of life;

and (5) eligible studies were randomized controlled trials, quasi-experimental trials, and pilot studies.”

 Protocol and registration: While it's mentioned that the review was registered with PROSPERO, it's not indicated if the systematic review protocol is publicly available for other researchers to consult.

Response: we have added information about the registration register available in prospero as suggested, please see pag 2

 Eligibility criteria: While eligibility criteria are provided, it may not be clear how these criteria were defined and if they were predetermined prior to the review.

Response: as previously suggested we have expanded the description of each component of PICOS strategy to make clear the eligibility criteria of each study to be included. Please see pag 3.

 Study selection process: It's mentioned that two researchers carried out the search, and discrepancies were resolved with a third researcher. However, there is no detailed explanation of how these discrepancies were exactly handled or if any specific method or tool was used for the selection process.

Response: we agree about the importance of methods section, and we have revised the information about how the selection process was performed to be sure that it is easy to understand.

8º Data extraction: Although it's mentioned that data extraction was carried out using predefined categories, there's no specification about what these categories are and how they were designed.

Response: we have added a clarification about the data extracted in our methods section.

 Outcome measures: There should be specifics about which primary and secondary outcome measures were considered before starting the review.

Response: we agree about the importance of methods section, and we have revised the information in our PICOS strategy.

10º Search in other sources: Although references of relevant reviews were checked, it's not clear if other sources, like clinical trial registers, were searched to identify unpublished or ongoing studies.

Response: we agree about the importance of methods section, and we have revised the information in our PICOS strategy to make it more specific.

11º Quality assessment and risk of bias: While the tools used to assess quality and risk of bias are described, there's no detailed description of the process nor how the results were interpreted and used in the analysis.

Response: we agree about the importance of methods section, and we have revised the information in our methods to make clearer the quality assessment and risk of bias.

12º Data synthesis: Random-effects and fixed-effect models are mentioned concerning heterogeneity, but it's not specified how it was decided which model to use in case of insignificant heterogeneity.

Response: we agree about the importance of methods section, and we have revised the information in our methods to make clearer decision in case of insignificant heterogeneity. Please see pag 3 and 4.

13º Additional meta-analysis: It's not clear if other types of analyses, like subgroup analysis or meta-analysis of individual studies, were considered.

Response: we agree about the importance of methods section, and we have revised the information in our methods to make it clearer. Please see pag 3 and 4.

14º Quality of evidence assessment: No specific tool or method (e.g., GRADE) is mentioned to assess the overall quality of the evidence for each significant outcome.

Response: we agree about the importance of methods section, and we have revised the information in our methods and added an adequate description of GRADE tool that we have included in our results section. Please see pag 3 and 4.

15º We recommend incorporating the GRADE (Grading of Recommendations, Assessment, Development, and Evaluations) approach in your systematic review process. This will not only enhance the credibility and clarity of your review but will also provide a systematic method for evaluating the overall quality of evidence for each significant outcome. Implementing GRADE can offer a more structured assessment and clear interpretation of the strengths and limitations of the evidence, thereby adding value to your conclusions and recommendations.

Response: we agree about the interest of using GRADE, and we have added an adequate description of GRADE tool, and we have included changes in our results, discussion and conclusion section.

16º We recommend that within the results section, specifically under the 'Selected Studies' subsection, you provide a detailed explanation of the flow diagram that clearly illustrates the study selection process. This diagram should break down the total number of studies identified, the number of studies excluded with specific reasons, and the studies that were ultimately included in the systematic review and those that were meta-analyzed. It's essential that the flow diagram clearly differentiates between the 12 studies that proceeded to qualitative analysis and those selected for meta-analysis. Furthermore, for the 38 articles that reported on exclusion, it would be beneficial for the flow diagram to detail the specific reasons for exclusion for each one, thereby providing greater transparency and understanding of the selection process.

Response: we agree about the interest of making results more complete, and we have revised the section to include relevant information.

17º After a detailed review of the corrections and responses provided in relation to the comments from the first review, we have noticed that some of the recommendations and suggested changes have been overlooked or ignored, particularly in the results section.

I would like to especially highlight the article by Kataoka K, et al., titled "Long-term effect of pulmonary rehabilitation in idiopathic pulmonary fibrosis: a randomized controlled trial." Although you mention that this article does not meet your search criteria, we would like to emphasize that it seems to align perfectly with your PICOS strategy. This study is conducted in patients with idiopathic pulmonary fibrosis, the intervention is pulmonary rehabilitation, there is a control group, the variable 6MWD that you consider is analyzed, and it is a clinical trial. Given this context, could you specify the exact reason for its exclusion?

We understand that conducting systematic reviews is a complex process, but it is essential to ensure the inclusion of all relevant studies to guarantee the validity and robustness of your findings. We urge you to reconsider the inclusion of this article and to reformulate the results section accordingly.

Response: we agree with the reviewer about the possible need for clarification in methods section, and in this line, we have deeply modified it. In this line, we think that now is easy to understand the exclusion of some studies. When checking the PICOS strategy the readers can understand what kind of design, population or intervention was not included, being now easy to understand why Kataoka was excluded. In this specific case, Kataoka et cols developed a program that does not include chest physiotherapy, and our intervention criteria for PICOS strategy was not respected.

18º Among other overlooked aspects from the first review, you failed to indicate whether the included studies received funding, provide a detailed explanation regarding the risk of bias in the selected studies, offer an in-depth description within the text of the tables presented, and give a comprehensive account of the results from each of the chosen studies. It should be noted that simply reporting doesn't suffice; a concise summary outlining intra-group and inter-group changes is necessary, among other considerations.

It's paramount that you closely review the feedback provided in the initial review and ensure strict adherence to all the specified recommendations.

Response: we agree about the interest of making results more complete, and we have revised the section to include relevant information.

19º Discussion Section:

- Clinical Relevance and Contextualization: Within your discussion, it would be beneficial to provide a deeper context at the outset that emphasizes the clinical significance of chest physiotherapy in patients with idiopathic pulmonary fibrosis. By focusing on the relevance of the intervention, readers can better grasp the potential impact of the findings on actual medical practice. Why is it crucial to study chest physiotherapy specifically for these patients, and what is its potential clinical significance?

Response: we agree about the interest of making discussion more complete, and we have revised the section to include relevant information.

- Exploration of Controversies: You mentioned that the results in previous literature are controversial. It would be valuable if a segment of the discussion delved deeper into these controversies. Are there methodological, demographic, or contextual factors that might explain these divergent findings? A detailed exploration of these aspects would enrich the discussion and offer a more comprehensive perspective on the current state of research in this field.

Response: we agree about the interest of making discussion more complete, and with GRADE recommendations added to our results we think that we can provide additional explanations for discussion. In this line, we have revised that section.

- Psycho-emotional Outcomes and Their Impact: While the study touches upon psycho-emotional improvements, I'd suggest delving deeper into the discussion regarding the relevance of these improvements. How do these improvements correlate with the patients' overall quality of life? Are there other studies supporting these findings? By contextualizing and delving deeper into these results, you can provide a more holistic view of the impact of chest physiotherapy on the overall well-being of idiopathic pulmonary fibrosis patients.

Response: we agree about the interest of making discussion more complete. In this line, we have revised that section.

Despite our initial feedback pointing out various grammatical and language errors, we still observe some persistent issues. We strongly recommend that you obtain a thorough review by a language expert, and attach a proofreading report with your revised manuscript. This will ensure clarity, format consistency, and the elimination of grammatical errors, thus enhancing the overall quality of the submission.

Response: we have asked previously for an English language proofreading.
